# Real-World Data and Machine Learning to Predict Cardiac Amyloidosis

**DOI:** 10.3390/ijerph18030908

**Published:** 2021-01-21

**Authors:** Elena García-García, Gracia María González-Romero, Encarna M. Martín-Pérez, Enrique de Dios Zapata Cornejo, Gema Escobar-Aguilar, Marlon Félix Cárdenas Bonnet

**Affiliations:** 1Fundación San Juan de Dios, Centro CC de la Salud San Rafael, Universidad Nebrija, 28036 Madrid, Spain; egarciga@nebrija.es (E.G.-G.); gracia.mgr@gmail.com (G.M.G.-R.); 2Hospital San Juan de Dios de León, 24010 León, Spain; encmartin@gmail.com; 3Sopra Steria, 28050 Madrid, Spain; enrique.zapata@soprasteria.com (E.d.D.Z.C.); mf.cardenas@soprasteria.com (M.F.C.B.); 4Escuela Superior de Ingeniería y Tecnología, Universidad Internacional de la Rioja (UNIR), 26006 Logroño, Spain

**Keywords:** artificial intelligence, real-world data (RWD), cardiac amyloidosis, heart failure, machine learning, predictive models

## Abstract

(1) Background: Cardiac amyloidosis or “stiff heart syndrome” is a rare condition that occurs when amyloid deposits occupy the heart muscle. Many patients suffer from it and fail to receive a timely diagnosis mainly because the disease is a rare form of restrictive cardiomyopathy that is difficult to diagnose, often associated with a poor prognosis. This research analyses the characteristics of this pathology and proposes a statistical learning algorithm that helps to detect the disease. (2) Methods: The hospitalization clinical (medical and nursing ones) records used for this study are the basis of the learning and training techniques of the algorithm. The approach consisted of using the information generated by the patients in each admission and discharge episode and treating it as data vectors to facilitate their aggregation. The large volume of clinical histories implied a high dimensionality of the data, and the lack of diagnosis led to a severe class imbalance caused by the low prevalence of the disease. (3) Results: Although there are few patients with amyloidosis in this study, the proposed approach demonstrates that it is possible to learn from clinical records despite the lack of data. In the validation phase, the algorithm first acted on data from the general study population. It then was applied to a sample of patients diagnosed with heart failure. The results revealed that the algorithm detects disease when data vectors profile each disease episode. (4) Conclusions: The prediction levels showed that this technique could be useful in screening processes on a specific population to detect the disease.

## 1. Introduction

Amyloidosis is a group of diseases characterized by the uncontrolled deposition of amyloid protein affecting multiple organs. This disease, in 50–65% of cases, attacks the heart causing an eventual increase in morbidity and mortality of patients. Among the more than 30 proteins that can make up amyloidosis, light chain monoclonal (AL) and transthyretin (ATTR) are the most common [1,2,3,4,5,6].

ATTR is produced mainly by the liver and functions as a transporter of thyroxine and retinol. Additional production of these compounds takes place within the choroid plexus of the brain and the pigment epithelium of the retina. The heart and the peripheral nervous system are the main places where the deposits accumulate, as well as the skin.

The accumulation is the cause of ATTR and organic dysfunction [2]. Autopsies have shown that 25% of people over the age of 80 have their myocardium infiltrated by amyloid TTR deposits. ATTR leads to cardiac amyloid deposition in patients after the sixth decade of life.

Cardiac amyloidosis (CA) is a rare form of restrictive cardiomyopathy that is difficult to diagnose [4]. Annually in the US, there is a prevalence of 5 to 13 cases per million population [7]. However, the actual prevalence of CA is not entirely clear because the disease is poorly understood, and its initial symptoms are nonspecific [7,8].

The difficulty of diagnosis leads to the use of clinical suspicion [8], causing many patients to die without adequate treatment or even a diagnosis. Detecting the probability that patients may suffer from amyloidosis is crucial when applying for therapeutic advances and improving the prognosis [9].

Clinical results depend on the extent of tissue affected and the type of amyloid fiber deposited. The suspicion of CA should relapse in patients presenting heart failure with preserved ejection fraction, unjustified left ventricular hypertrophy and multiorgan involvement, being this the most common cause of restrictive cardiomyopathy. In 20% of the cases, there are predominant cardiac symptoms, and in 5% of them, only isolated cardiac involvement occurs. In these cases, it is common that in the vast majority of patients, the involvement occurs in more than one organ [1,2,5,10].

In the elderly population, heart failure (HF) is a common condition. For both doctors and healthcare providers, the diagnosis is simple when it comes to common types of HF. The less common types are those that delay diagnosis or even fail to be identified and are associated with rare diseases such as CA [6].

Early diagnosis and characterization of CA are necessary for many patients, but the above factors only contribute to the delay of such diagnosis. TTR amyloidosis has a low diagnostic rate, and the symptoms the patient displays are a treatable cause of heart failure [1]. Only 13% of ATTR cases are detected in heart failure patients with preserved ejection fraction. Myocardial involvement is the most important driver of the prognosis of systemic amyloidosis. Stratification of patients, in turn, is essential for accurate medical or surgical prognosis and treatment [10].

Medical diagnoses are generally based on patterns learned during clinical practice, based on individual experience about patients, losing the advantage that technology offers by analyzing data according to latent knowledge and patterns present in the data recorded in electronic medical records [11].

The starting point of personalized medicine must be an accurate diagnosis, which requires analysis beyond the data coded in the systems used in clinical practice [12,13,14,15,16]. The introduction of electronic health records (EHRs) has sparked widespread interest among clinical researchers in this field, as the complete and extensive use of clinical datasets offers great potential for transforming the healthcare system [11].

Nursing records, for example, provide information about patient conditions, with the data often reflecting nurses’ cumulative experiences in recognizing patient deterioration through the interpretation of physiological and psychological signs. Previous studies used the frequency of free-text comments as an indicator of nursing concern about the patient, and the frequency of comments was generally associated with increases in cardiac arrest and mortality [8].

In the development of diagnostic algorithms based on machine learning, common data extraction strategies identify terms that are manually preselected by physicians, sometimes supplemented with additional related vocabulary [17,18,19,20]. However, another way to obtain information from the written text is by processing natural language through subject models. This strategy is based on the identification of concurrent words with the aim of determining the latent semantics of the topics within the text. It has been demonstrated that in this way, significant notions can be obtained from a group of notes in the medical records of an intensive care unit, which were used to classify patients at risk of mortality with high precision [21].

Thus, artificial intelligence processing tools can close the gap between the large amount of data generated and the limited cognitive capacity of the human mind [22,23].

The aim of this study was, on one hand, to process data from electronic clinical records in order to characterize the patient’s health status by means of a numerical vector, which will secondly allow training an automatic learning algorithm that facilitates the early identification of CA patients.

## 2. Materials and Methods

### 2.1. Design

Retrospective Transversal Descriptive Observational Study.

### 2.2. Participants

We analyzed and processed ten years of anonymized electronic clinical records (both medical and nursing) (2009–2019), which belong to 11,586 patients over 65 years of age, with an average age of 87 ± 17 and a roughly balanced sex distribution of 58% women and 42% men. The total results in 16,620 episodes of medical consultations. The data analyzed include structured and unstructured information. One characteristic of unstructured data is that they collect free-text notes that doctors and nurses have left about the patient’s process.

Patient inclusion criteria: we selected subjects older than 65 years as TTR amyloidosis symptoms appear after the age of 65. To analyze all patients who may be admitted for another pathology, and not only in the cardiology and internal medicine services, we considered including the total number of patients admitted to the hospital during the last five years. In this way, possible undiagnosed cases would be susceptible to detection. By including all the records, both structured and unstructured, it was possible to analyze the primary diagnoses and comorbidities, detecting the weight that these could have had on identifying the patient with real CA.

Patient exclusion criteria: oncological patients with active treatment.

### 2.3. Features Extraction

In selecting characteristics, sociodemographic variables, clinical evolution variables, and other variables collected in the electronic clinical record in a structured way by doctors and nursing personnel were included, such as clinical variables derived from the evaluation, complementary tests, diagnosis, and prescribed treatment.

Medical diagnoses were collected through the International Classification of Diseases (ICD-9 and ICD-10), and the variables corresponding to the nursing care process through the NANDA (North American Nursing Diagnosis Association) NIC (Nursing Interventions Classification)-NOC (Nursing Outcomes Classification) taxonomy. Variables extracted from the relevant clinical studies in the field of cardiac amyloidosis were added.

Unstructured data were collected from the clinical reports, extracting from the free text fields of the electronic clinical records the text written by the doctors and nurses in the clinical evolutions during the entire hospitalization process. As these are free text fields, it can be problematic when analyzing the health care records due to the lack of structure. However, we believe that this analysis may be the key to identify red flags to detect possible undiagnosed CA cases. In order to identify the most commonly written terms in the evolutionary ones that could behave as red flags, two expert sessions were held in which a total of 8 geriatricians, internists, cardiologists, pharmacists and nurses participated.

#### 2.3.1. Data Processing

Data analysis was initiated by searching for diagnosed cases of CA in extracted medical records. Twenty-six patients with amyloidosis were detected according to ICD 9 and ICD 10 coding (depending on the system used in the temporary sections by the hospital). From these profiles, the construction of the specific characteristics vector began. Figure 1 shows the most frequent diagnoses of the group of patients with CA.

To construct the vector of characteristics of the patient with amyloidosis, the process described in Figure 2 was carried out. The figure shows a summary of the entire construction process of the training data set. As shown in the upper left corner of Figure 2 (step 1), we loaded the clinical records into a data dictionary with quick access to memory for manipulation.

The data went through a preliminary preprocessing stage (step 2) before new fields were generated, and the transformation pipelines were executed. The cleaning process removed white spaces, special characters, null values and columns or rows with constant values. In addition, in order to join the data tables coming from different sources, a new episode primary key field was created. Once the information was consolidated, new fields were added (step 3). These fields were calculated following specific characteristics of the CA [24,25] that, according to the experts, may detect the disease [26,27,28]. In step 4, we discarded many fields with low-quality data, reducing memory and unnecessary operations. As shown in Figure 2 (step 5), the processing of each type of variable (text/categorical/numeric) was carried out separately to be treated in a particular way according to their nature.

The pipelines are programming artifacts that define a process that can be reproduced with new data and can even be parallelized for faster execution. When pipelines end their execution, the training variables are obtained. Finally, we joined tables by episode key in a trainable matrix/table (step 6).

#### 2.3.2. Categorical Data, Generation of New Characteristics and Categorical Pipeline

We review categorical data processing that we can examine in two parts: the categorical variables generated in the feature engineering and the variables generated in the categorical pipeline.

In (step 3), new variables were created manually following the clues provided by clinical practice. In this stage of characteristics engineering, the medical diagnosis codes were used, specifically the ICD-9 and ICD-10 codes. In this same line, a selection of crucial diagnoses was made, and the equivalence between both versions was mapped. Table 1 shows a summary of the information that was included in the study. The medical and nursing diagnoses included in the study were extracted from two working sessions with experts in which signs and symptoms frequently detected in clinical practice and which could be masking a diagnosis of cardiac amyloidosis in a patient admitted to the hospital were discussed.

This stage’s results made it possible to describe potential patients with CA, using indicators that alerted them about the disease. In the same way, it was possible to know their general health status by analyzing the comorbidity of other common diseases. As shown in Table 2, the nursing care codes identified in the expert session were used.

Finally, as mentioned, the other part of categorical data were generated in the pipeline, resulting in an automatic dummy encoder with a custom categories dictionary. A generic dictionary was elaborated to facilitate the reproducibility of encoded variables in the categorical pipeline (step 6), where all possible categories of each field were annotated for later dummy encoding.

This last step ensures that all participants in the analysis will be aware of any coding applied to the data and that the same categorical variables will be created from each field in the encoder.

Categorical data are widely used, and their relevance to analyzing the services provided to the patient should be highlighted. They have generally been used to represent vital signs, nursing care, diets, pharmacy, and other medical services (see Table 3, categorical variable).

#### 2.3.3. Treatment of Numerical Information, Numerical Pipeline

We start explaining the numerical pipeline process as follows: the numerical fields describing vital constants, glucose levels, and the hydro electrolytic balance were selected. New statistical variables were calculated using numerical fields to characterize these values’ distribution along with health care. New variables are the total of records, the arithmetic mean, the variance, the minimum value, the maximum value, and the percentiles 25, 50, and 75%. The meaning of our numerical fields is summarized in Table 3 (numerical variable).

#### 2.3.4. Treatment of Unstructured Information, Text Pipeline

Unstructured text is doctors’ and nurses’ notes with information describing various relevant events during hospitalization. Topics treated in our data are summarized in Table 3 (Textual variable). In these free text variables, natural language processing techniques have been used to standardize the text of the annotations into a series of numerical vectors. This set of vectors is commonly known as a bag of words. The first step is to generalize the text by converting all texts to lowercase and then removing stop words. It was necessary to use a translation of abbreviations with medical dictionaries and a handwritten reference database of the nursing staff.

The cleaned-up texts were put into the token and then reduced to their root. Due to computational capacity limitations, it was necessary to define a maximum number of vocabulary and establish the documents’ minimum frequency of appearance. This process allowed the creation of a vector for each document found in the clinical records and adding all the documents generated by each patient who visited the hospital during the study period.

In summary, 83 new fields were calculated manually with the help of healthcare professionals (see Table 1 and Table 2). The pipeline results were 49 variables from numeric fields, 6928 variables from categorical fields and 643 variables from text fields. Table 3 shows other data used to describe the patient’s health and care status worldwide from a real-world data source.

The final aggregation steps (steps 5 and 6) generated a matrix suitable for machine learning training. The matrix has 7620 columns or variables and 16,620 rows, as many rows as the sum of all patient’s stays (episodes). Each row is what we call an episode vector.

We have discussed in detail the proper data preparation suitable for the training phase of the models. In the following steps, we will examine the training process and describe the considerations used to design the most generalizable model possible.

### 2.4. Training and Algorithm Design

#### 2.4.1. Validation Techniques

The cross-validation [29] took into account five stratified folds (k = 5), where each fold affected the training in 4/5 of the data and the validation in the remaining 1/5. Stratification ensures that the ratio of 0’s and 1’s is the same in the training and validation sets. The algorithm learns using all the data and different patients each time. This mechanism is useful when the objective diagnosis is scarce.

Cross-validation proves useful in working with scarce data. The model selection phase is verified that other complex models such as a random forest or neural networks show signs of overfitting. Specifically, with these models, zero (0) true positives are obtained in several folds of the cross-validation.

For example, a neural network (two hidden layers structure with 100 and 10 neurons, respectively, having a rule activation function and a cross-entropy loss function, with an Adam solver) gets only 0.6 ± 0.5 true positives. While in the training set, zero (0) false-negatives and zero (0) false-positives are obtained. Such behavior indicates memorization of the training set throughout the five validation folds and discourages the use of complex models prone to overfitting.

The metrics’ behavior and the cross-validation confounding matrix showed the robustness of the model highlighting, low dispersion of results, and an adequate generalization. The model that fits best is a logistic regression in which the modeling between predictor variables and the prediction of class 0: not sick, 1: sick is achieved efficiently with a sigmoid function.

#### 2.4.2. Ending Algorithm

After an iterative process, the operations and transformations of the proposed algorithm steps are defined as follows: (1) Imputation of the missing values with the mean; (2) selection of variables with a variance greater than zero; (3) normalization of variables to [0.1] required to apply principal component analysis (PCA); (4) application of a PCA covering 99% of the variance (reducing a high number of features successfully); (5) standardization of variables; (6) apply an undersampling using the centroids of each cluster (120 zeros [0] about 30 ones [1] per fold); (7) apply a slight oversampling with a synthetic minority oversampling technique (SMOTE) [29] of 120 zeros [0] and about 60 ones [1]. (8) Finally, the training with the logistic regression.

#### 2.4.3. Metrics for Evaluating Results

Sensitivity, specificity and accuracy are appropriate metrics to characterize the performance of the proposed training tests. To evaluate the model’s learning ability in class imbalance scenarios [30], the area under the receiver operating characteristic curve (ROC) was included. Disease prevalence affects sensitivity, specificity and accuracy [31]. The latter is relevant for less prevalent diseases. F1, on the other hand, besides being very sensitive to improvement or worsening of results, allows measurement of success in the positive class and is timely in cases of unbalanced learning.

Once the final algorithm was developed, training was done initially with the general sample and then with a sample of patients diagnosed with heart failure.

## 3. Results

After reviewing the training approach, the next step was to look for the diagnoses most relevant to the prediction. On one hand, the model trained on the whole population or all clinical records. On the other hand, a sample of heart failure patients was used for training. In this sample, there are 2861 patients with heart failure and 27 with CA.

This sample yielded 3806 episodes, equivalent to an unbalanced ratio of one (1) positive: 89 negatives. When examining the confounding matrices of the validation sets of both training pieces (see Table 4 and Table 5), the evidence showed that in both test scenarios, some episodes of successful patients were detected and that the number of false-negatives was low. A problem with the first approach is the number of false positives.

The results of training with the sample of heart failure patients are shown in Table 5. The number of true positives is higher than false-negatives. The proportion of false-positives, on the other hand, decreases markedly while the number of detected episodes increases.

Table 6 shows a comparison of sensitivity, specificity, accuracy, area under the ROC curve and F1. For patients with heart failure, the ROC area 0.88 ± 0.08 shows a good result for learning the model. As shown in Figure 3 and Figure 4, the improvement in the models’ training when using patients with heart failure is noticeable.

When working with the entire dataset, the sensitivity, accuracy and F1 metrics are low and could be significantly improved if there were more CA cases. With very low disease prevalence, sensitivity and accuracy decrease even though specificity remains high. This result can be justified with a Bayes’ theorem analysis applied to the result.

On the other hand, when working with heart failure patients, an increase in sensitivity of 0.56 ± 0.19 is obtained while maintaining high specificity. This increase suggests that the model could be useful in screening tests or health campaigns for CA follow-up.

The benefit is justified in the gain curves in Figure 5 and Figure 6. Figure 5 shows that with 20% screening, the proposed algorithm can potentially locate around 50% of episodes of patients with CA. This screening would avoid examining 30% of the total population. The result is a moderate improvement over a random selection of patients (represented by the diagonal from the lower left to the upper right corner).

Figure 6 shows that screening 20% of the population with heart failure could find 80% of the episodes of patients with CA. This would avoid screening 80% of the episodes of heart failure patients. The drawback of using this approach is that it limits the possibilities of early detection as patients have considerable heart involvement.

## 4. Discussion

In the context of chronicity and aging, efficiently managing health expenditures is a necessity. The structured data registered in the clinical records do not explain the processes by themselves; however, the unstructured information, written by doctors and nurses, may be the key to knowing about the progress of the diseases. Therefore, this project will analyze clinical records’ open text as an essential part of obtaining variables to be introduced in the final algorithm modeling. The application of machine learning in the interpretation of doctor’s and nurse’s language and the coding and the correlation of variables to each other by proximity is a challenge with a potentially significant impact on clinical decision-making [14].

There is currently a problem of underdiagnoses of Cardiac amyloidosis, which involves the participation of other organs, loss of quality of life, and the possibility of benefiting from adequate treatment [2,3,4,8]. Valuing a predictive model that facilitates a differential diagnosis, quickly and practically, of patients with cardiac amyloidosis will allow therapeutic objectives to be adequately addressed [6,12].

Patients with heart failure associated with CA show indistinguishable symptoms from those reported by patients with other phenotypes of heart failure. Moreover, the classic diagnostic tools commonly used to diagnose heart failure (cardiac biomarkers, electrocardiographic markers, imaging techniques) require advanced expertise to perform the differential diagnosis between amyloid and non-associated heart failure. The inclusion in this stage of predictive algorithms based on advanced artificial intelligence techniques can be useful in clinical diagnosis [6].

Predictive algorithms can be used as a population screening system for a specific disease, in this case, CA, with the aim of diagnosing patients who suffer from the disease but do not have a correct diagnosis. In this way, clinical management could be optimized, improving the clinical, economic and social impact on the patient. On the other hand, the identification of red flags and keywords identified in the algorithm can be used in intelligent support systems in the electronic medical record, so that when the doctor or nurse write in the clinical evolutions, recommendations are made based on the appearance of any of those keywords in the text. These systems would support decision-making with a direct impact on clinical care. Algorithms cannot establish a cause–effect relationship and, therefore, cannot replace the physician in clinical practice; however, machine learning can recognize numerical patterns that emerge from large volumes of health data and are not easily recognizable by humans.

Artificial intelligence (AI) has made significant contributions to the healthcare industry; however, its effect on medical diagnosis has not been well explored and validated in clinical practice [32]. Harada et al. [33] published an essay comparing the thought process between a computer and a diagnostician. The essay concluded that machine input information, applying AI, could not be weighted in order of diagnostic significance. The comorbidities, patient context, and disease temporality were difficult to detect and analyze. In our study, these difficulties were contemplated, planning to minimize them by having clinical experts and data scientists meet periodically during the analysis process. Knowledge of the disease, interpretation of the results, and guidelines for AI are fundamental in achieving AI implementation as a support for clinical decision-making. Conducting studies like this one, in which it is possible to work with real data, is a necessary step and a good starting point to reach these solutions.

Another relevant point of this type of work is to handle with prudence the interpretability of the models. In this project, it is necessary to go deeper into the meaning and importance of the variables obtained after reducing dimensions. Health experts should take the time to analyze the list of variables that emerged from the exploration of related diagnoses and nursing care and patient services. Analysis of these variables may even yield new disease indicators or eliminate those that may be causing the noise. Clinical experts in the diagnostic process, helping the data scientists in the decision-making during the applied analysis, should validate the applied AI tools’ findings.

It should also be noted that the most common types of amyloidosis are immunoglobulin light chain (AL) and amyloid transthyretin (ATTR) amyloidosis. In AL, the treatment is based on chemotherapy and/or stem cells. In this study, it was not possible to distinguish both types of amyloidosis in the patients identified with CA, so it was decided to exclude patients with cancer treatment from the study. However, it would be of interest for future studies to do subgroup analyses, including the different types of amyloidosis [34].

Tran et al. [32] used an approach in their study that combined bibliometric analysis with a more complex analysis of abstract content through exploratory factor analysis and latent Dirichlet assignment. The study revealed new areas and emerging themes in research focused on early and population-specific stroke and heart disease detection. In light of these findings, the authors suggest that to maximize the benefits of AI in detecting heart disease, one should delve into currently under-researched issues such as data management, the reliability of the AI model, and the validation of its clinical utility.

Models have been trained despite the initial difficulties of the classification problem. The prediction yielded better results with the sample of heart failure and CA patients. With a sensitivity and specificity of 0.56 and 0.96, respectively, the result is comparable to other screening tests present in the literature [6,14]. The low accuracy and modest F1 can be explained by the disease’s low prevalence [35]. New data may be incorporated into this proposal to see improvements in the results.

Based on the latest advances in data science and applications of machine-learning-based algorithms, we have hypothesized that patients with heart failure caused by amyloid deposition may show different clinical patterns than patients with the same diagnosis without association with CA. These patterns, in both structured and unstructured data, we believe are detectable by intelligent statistical approaches. Other authors, such as Agibetov et al. [6], used the same approach using the biomarkers that are routinely and widely used in the diagnosis of patients with heart disease, obtaining model results with 0.75 ROC AUC with sensitivity 84.6%, specificity 71.7%, positive predictive value 47.1%, and negative predictive value 96.6% (FOR 3.4%).

With the extracted clinical data, both structured and unstructured, it has been possible to train models and obtain a result despite the scarcity of positive disease cases in the data set. Research by Basharat et al. [14] in bioinformatics has already yielded results favorable to the use of the unstructured text of clinical records recorded by physicians and nurses in the generation of diagnostic algorithms in cardiac patients. The validation of the model has been as demanding as possible. The model has always been validated on samples with representative ratios of the real prevalence of the population. In other words, the resampling techniques are not applied to the validation sets. In addition, the cross-validation helped to keep away the risks of an over-adjustment to the data set or the influence of dependent samples on the validation. This can be seen in the small deviation of metric values in the different sets. In other words, the results tend to be consistent across all five-validation sets. Thanks to the results of the validation and to the techniques applied, it is considered that the performance of the model obtained is difficult to improve with the available data.

The most favorable results were obtained with a sample of patients with heart failure and stroke. With a sensitivity and specificity of 0.56 and 0.96, respectively, the result is comparable to that of other screening tests present in the literature [30,31]. The low precision values and the F1 can be explained by the low prevalence of the disease, although they should be the subject of future improvement efforts. The model’s predictive ability and learning have been excellent, as shown by the ROC curve with an area of 0.88 in the best case. The results promise to introduce more data from unstructured information such as medical, radiological, nursing reports and diagnostic tests.

The work published by Agibetov et al. [6] demonstrates that machine learning applied to the analysis of basic laboratory parameters is useful to generate a profile of patients with heart failure (H) related to cardiac amyloidosis (CA). Compared to non-CA H patients, opening a potential new avenue in the diagnosis of CA that allows clinicians support in clinical decision-making.

The treatment of data by health care episodes is a novel approach that favors the detection of rare diseases. In this approach, the patient himself generates information with sufficient variance to be considered different in the training of an automatic learning model. The high dimensional vectors generated from large data sets to define a patient’s stay in a healthcare facility can help quantitatively define a patient’s health status and care. Improving the collection and interpretation of these vectors could help discover new patterns that improve patients’ quality of life or the efficiency of the system. In this process of collecting quality data and its subsequent exploitation, administrators, medical and nursing staff, scientists, and engineers’ multidisciplinary commitment is fundamental [36]. The possibilities are there; it all depends on the quantity, quality, ease of processing and access to data.

The final test that the algorithm must pass is the model’s production; for this, the result here exposed must be generalizable out of the data set. Verifying that this generalization occurs should be the work of additional tests and studies. Another future possibility arises from the fact that the number of false positives is small enough to carry out a clinical follow-up. Verifying whether these false positives are undiagnosed patients will validate the model’s predictive capacity in the original data set.

## 5. Conclusions

The processing of information by “health care episodes” is a novel approach to the problem of detection of rare diseases. The patient himself generates information with sufficient variance to be considered different in the training of an automatic learning model.

High dimensionality vectors created from large volumes of data allow descriptions of the patient’s stay in hospital. Vectors also quantitatively characterize the patient’s health status, nursing care and services provided. Improving the construction and interpretation of these vectors can help discover new patterns that meet patient needs.

The incorporation of unstructured data, of the temporality of the disease, and the identification of flagship networks from scientific evidence and the consensus of clinical experts with experience in the field, in this case of geriatrics and cardiac pathology, are crucial to the development of successful diagnostic algorithms.

The use of artificial intelligence is considered very useful in supporting medical decision-making. However, it is necessary to externally validate the use of algorithms as a diagnostic test as well as the cost-effectiveness of its implementation. For this, prospective studies are necessary for which the effectiveness of the algorithm is tested, evaluating the care, economic and social impact. In the case of CA, it will be carried out in the next phase of the investigation.

## Figures and Tables

**Figure 1 ijerph-18-00908-f001:**
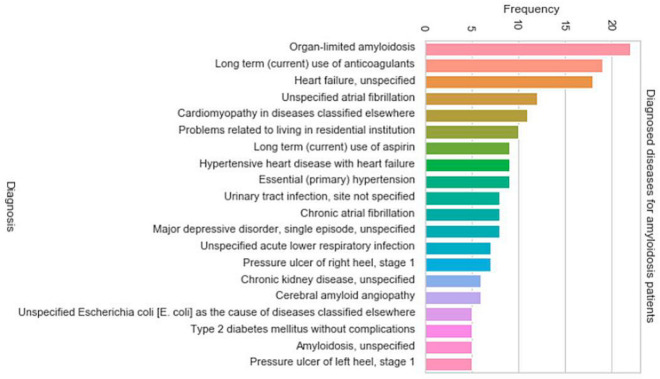
Frequent diagnoses of the group of patients with cardiac amyloidosis (CA).

**Figure 2 ijerph-18-00908-f002:**
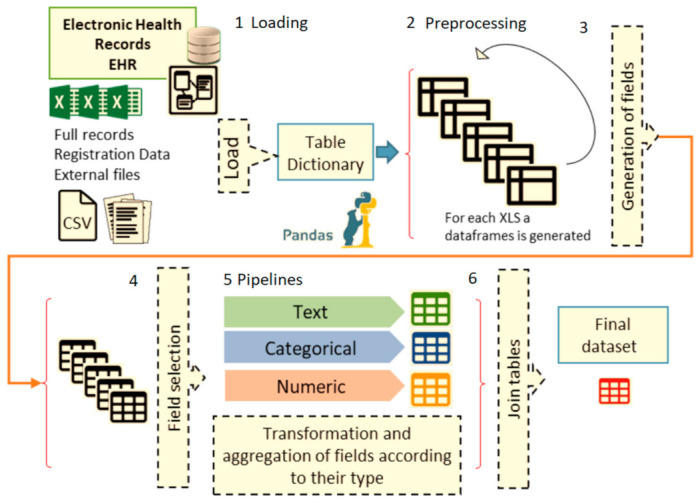
Stages of clinical data processing.

**Figure 3 ijerph-18-00908-f003:**
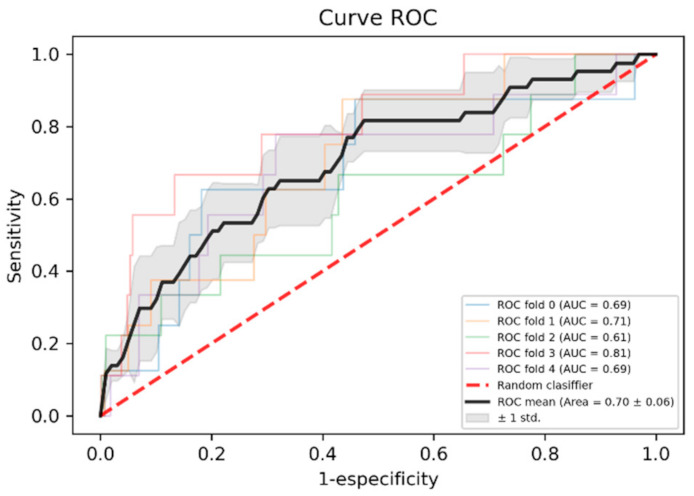
Receiver operating characteristic (ROC) curve of the model trained with the general population.

**Figure 4 ijerph-18-00908-f004:**
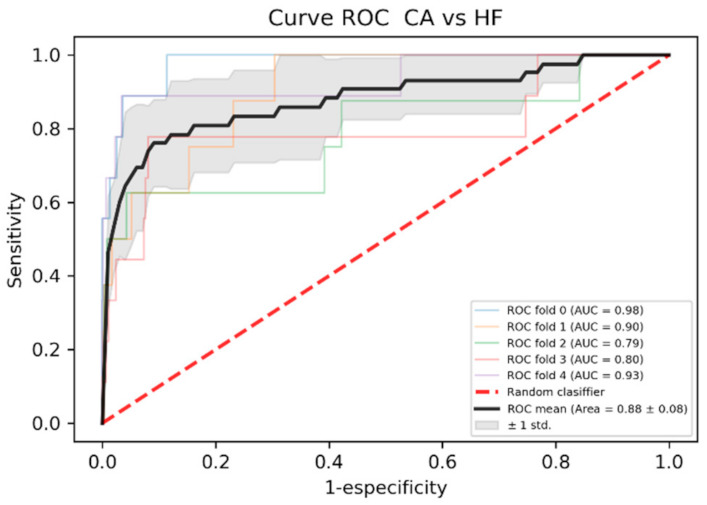
ROC curve of the model trained with the sample of patients with Cardiac Amyloidosis (CA) and Heart Failure (HF)**.**

**Figure 5 ijerph-18-00908-f005:**
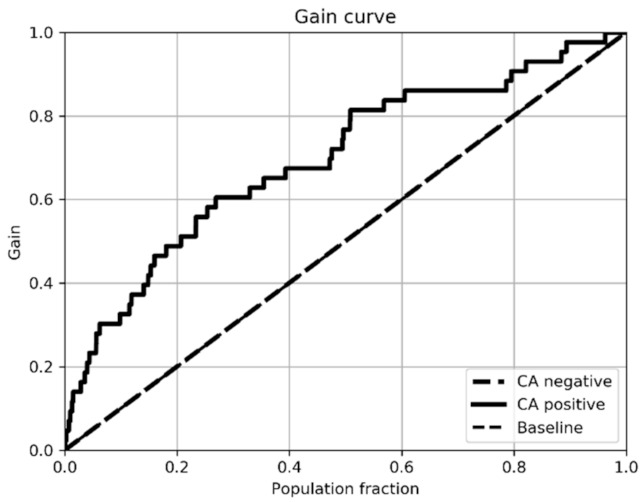
Cumulative gain curve of the trained model with the general population.

**Figure 6 ijerph-18-00908-f006:**
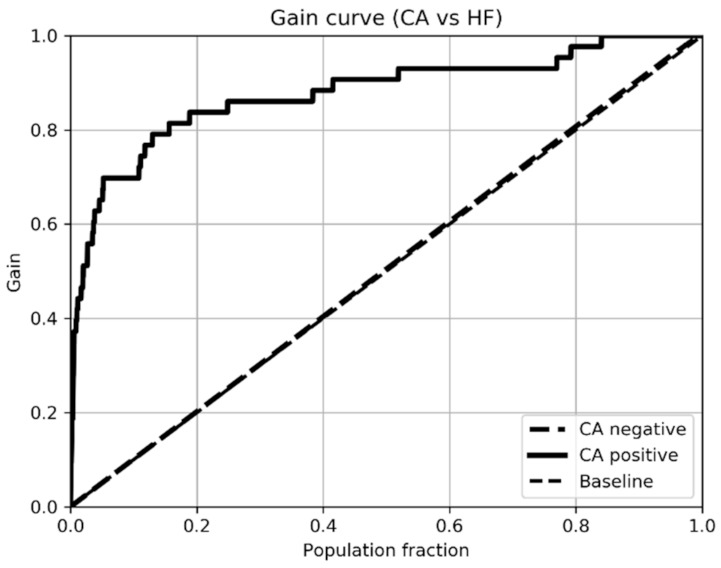
Cumulative gain curve of the trained model with the sample sample of patients with Cardiac Amyloidosis CA and Heart Failure HF.

**Table 1 ijerph-18-00908-t001:** Indicator diagnoses of cardiac amyloidosis obtained from ICD * coding.

Primary	Diagnose
Cardiovascular manifestation	Atrial fibrillation, arrhythmia, extrasystole, pacemaker, hypertension, heart failure, restrictive cardiomyopathy, acute pulmonary edema, pericardial effusion, cardiac tamponade.
Extracardiac manifestations	Multiple myeloma, monoclonal gammopathy, ventricular hypertrophy.
Central nervous system	Progressive dementia, headache, ataxia, seizures, spastic paresis, stroke.
Ocular manifestation	Intravitreal deposits, vitreous opacity, periorbital purpura, glaucoma.
Nephropathy	Proteinuria, kidney failure, chronic kidney disease.
Gastrointestinal manifestations	Weight loss
Peripheral neuropathy (sensory or motor)	Carpal tunnel syndrome, spinal canal stenosis.
Autonomic neuropathy	Bilateral sensory-motor polyneuropathy, dysautonomia, erectile dysfunction
Others (to characterize the patient)	
Common diseases of the elderly	Common cold, flu, gastroenteritis, allergy, arthritis, osteoarthritis, Alzheimer’s, dementia, Parkinson’s, etc.

* ICD: International Classification of Diseases (ICD9-ICD10).

**Table 2 ijerph-18-00908-t002:** Summary of nursing care extracted from the dataset.

Signs	Diagnose
Vital signs	Blood pressure, heart rate, temperature, oxygen, sugar
Nursing diagnoses	Constipation, subjective constipation, diarrhea, fecal incontinence, risk of constipation, impaired urinary output, urinary retention, nausea, or risk of falls.
Nursing signs	Aversion to eating, elimination of hard, dry and formed stools, minimum elimination of three stools/day, continuous flow of urine that occurs at unpredictable intervals, constant dribbling of loose stools, incontinence, report of intake below the recommended daily amounts, small and frequent urination, nausea, nausea and vomiting, diarrhea or overflowing of liquid stools.
Respiratory signs	Dyspnea, dyspnea on exertion, cough.
Nutritional signs	Aversion to eating, report intake lower than recommended daily amounts.
Abdominal signs	Abdominal bloating and pain.
Confusional signs of mental state	Increasing agitation, agitation, confusion/agitation, disorientation.
Instability	Impaired physical mobility, impaired ambulation, risk of falls.
Others	Confirmed care, dressings and bandages, manifestations (other signs/symptoms), assigned typical care plan, causes (coded etiology).

**Table 3 ijerph-18-00908-t003:** Overview of diagnostics by variable type.

Type	Content
Categorical	Diets, feeding routes (types), identification and administrative data (sex, age divided into percentiles, code of the situation at discharge, code for the reason for discharge), coded medical procedures, patient consultations (code of inter-consultation requested and carried out), prescription and maintenance of drugs and serums (code of administered drug), patient benefits.
Textual	Patient consultations (description of the reason for the consultation), patient medical orders (description of the medical order to the patient and observations), patient benefits (reason or observations), nursing assessments, medical assessments.
Numeric	Alteration of the water and electrolyte balance (number of evacuations) vital signs (minimum tension, maximum tension, temperature, oxygen saturation level, amount of blood glucose).

**Table 4 ijerph-18-00908-t004:** Results of the model trained with the global population.

Confusion Matrix		Prediction		
		Negative	Positive	Total Real
Real	Negative	3270.4 ± 18.64	45.0 ± 18.36	3315 ± 1
	Positive	7.0 ± 0.63	1.6 ± 1.02	9 ± 1

**Table 5 ijerph-18-00908-t005:** Results of the model trained with heart failure patients.

Confusion Matrix		Prediction		
		Negative *	Positive *	Total Real
Real	Negative	726.2 ± 2.79 ▼	26.4 ± 2.5 ▲	753 ± 1
	Positive	3.8 ± 1.72 ▼	4.8 ± 1.72 ▲	9 ± 1

* Increases the value (▲) or decreases the value (▼) with respect to the previous analysis (see Table 4).

**Table 6 ijerph-18-00908-t006:** Comparison of the results of the proposed training scenarios.

	Population	Heart Failure Patients
Sensitivity	0.18 ± 0.11	0.56 ± 0.19
Specificity	0.99 ± 0.01	0.96 ± 0.00
Accuracy	0.04 ± 0.02	0.15 ± 0.06
F1	0.06 ± 0.04	0.24 ± 0.09
ROC AUC	0.70 ± 0.06	0.88 ± 0.08

ROC AUC: Area Under the Receiver Operating Characteristic Curve from prediction scores.

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
