# Peer review of "Real-World Data and Machine Learning to Predict Cardiac Amyloidosis"

_ijerph, 2021, doi:10.3390/ijerph18030908_

Round 1
Reviewer 1 Report
The author aimed to analyse a new innovative method to help clinicians in diagnosis of cardiac Amyloidosis. They investigate clinical records. However some major issue need to be addressed
Major
- in methods it is not clear if they analysed medical or nursing records.
- They analyse patients aged over 65. Despite Amyloidosis is a disease that afflicts more the elders, it has to be indicated why they choose only geriatric population instead of all patients that come to the internal medicine/cardiology wards. Moreover, they have to indicate also the wards and hospital included in the analysis
- in the same section, paragraph 2.2 lacks in specific evaluation of the comorbidities patients are affected. Many patients over 65 have comorbidities that may influence the diagnosis. Artificial intelligence should be helpful in identifying real amylodosis especially when multiple confounding factors are present. The authors must specify how the population studied was afflicted by disease that may influence a cardiac hypertrophy more than Amyloidosis.
- In method is not specified how they check the diagnosis of amyloidosis. Moreover I wonder to know if all patients perform at least a periumbelical fat biopsy to confirm the diagnosis of amylodosis. Unfortunatelly, there is no biochemical marker that should be used to confirm diagnosis other than histology. Please address.
- There are no indication about the method used for cardiac evaluation. I guess the authors studied only the ICD code. If so, only a suspicious of cardiac amyloid involvement should be indicated. A statement about how they identify cardiac amyloidosis is mandatory.
- In results it is not indicated how machine learning is more effective than the clinical judgment in support identification of cardiac amyloidosis
- Some paragraphs are difficult to read and understand. I suggest an extensive language revision by a native speaker reviewer
Minor
- Cardiac Amyloidosis is reported with two different acronym: please check and address
- Numerical parameters need to present also a range or standard deviation (especially in methods)
Author Response
Please see the attachment
A point-by-point response to the reviewer’s comments can be found at the uploaded Word file.
The new main text of the manuscript have been revised and uploaded and changes are higlighted in red font.

Reviewer 2 Report
This manuscript investigates a machine learning application for Cardiac Amyloidosis, utilizing digital clinic records. The paper is well-written, with clear research motivation, methods, and findings. Yet, two major issues need to be addressed before publication.
First, I find the data sources, collection, and processing process are not sufficiently described. While the authors mentioned the sample size and age distribution in 2.2, there should be more explanations on data collection without disclosing sensitive information. Figure 1 is necessary but needs to be improved with more clarity. Mainly, the graphic components should directly reflect the sections described in the texts. Authors need to elaborate on the data process illustrated in Figure 1 beyond a very brief sentence in 2.3.1.
Second, the authors conclude that the results would be useful in screening processes on a specific population to detect cardiac amyloidosis, but there is little discussion on how the proposed machine learning algorithm can be implemented into the current screening process, particularly for cardiac amyloidosis. The authors provide complete explanations on the model output and performance comparing to previous studies in the discussion section. Still, there is a lack of strong reasoning to support why the machine learning predictive algorithm is especially valuable and necessary for cardiac amyloidosis screening. Thus, I suggest the authors elaborate on how machine learning and NLP address specific cardiac amyloidosis issues to highlight their methodological contribution.
Overall, this manuscript presents a valid research question, data processing and modeling methods, and real-world application. This study is suitable for the journal's aim and scope and appropriate for publication after minor revisions.
Author Response

(The authors gave the same response as above.)

Reviewer 3 Report
In this work, the authors analyze and characterize the cardiac amyloidosis disease and proposes a statistical learning algorithm that helps to detect the disease. The approach consisted in using the information generated by the patients in each admission and discharge episode and treating it as data vectors to facilitate their aggregation. The large volume of clinical histories implied a high dimensionality of the data and the lack of diagnosis led to a severe class imbalance caused by the low prevalence of the disease. Preliminary results demonstrated that it is possible to learn from medical records despite the lack of data. There are some minor issues which I encourage the authors to consider: 1. For the chosen variable, I think that more analysis is needed. Data mining techniques could support the variables selection and therefore, validates the proposed formulation. 2. There is an important lack of novelty. The formulation (based on correlation and regression techniques) is a little bit simple in my opinion and other approaches such as based on convolutional neuronal networks or even fuzzy logic could be more robust in terms of classification, repeatability and temporal stability. I think that more discussion about this issue could be a nice complement for the current manuscript. 3. There are some grammatical/style errors. A grammar/style revision has to be carried out before the manuscript can be considered for publication.Author Response
Please see the attachment
A point-by-point response to the reviewer’s comments can be found at the uploaded Word file.
The new main text of the manuscript have been revised and uploaded and changes are higlighted in red font.

Round 2
Reviewer 1 Report
The authors replayed to all the issue required. They improved the manuscript: it is more clearly explained the patients studied.
They indicated comorbidities were included in their machine learning process. I appreciate they exclude oncological patients due drug-related myocardial dysfunction is a great counfounder: in this initial proposal these patients may disorient the analysis and readers. However, I suggest to include a statement in discussion indicating that a specific analysis is needed.
On the contrary, arterial hypertension has been indicated in table I as an indicator for amyloidosis. I expected that arterial hypertension may be considered as possible confounding factor, at least. In fact, hypertensive cardiopathy is misleading for an initial amyloid deposition. Similarly, Diabetes is never mentioned but it is one of the most important confounding factor as diabetic cardiopathy. I suggest to include a table indicating comorbidities that should induce heart damage, starting from both hypertension and diabetes, stressing those which are counfounders. A clinician guess to find the weight of these comorbidities in the diagnosis of cardiac amyloidosis.
The authors also replay they check and solved the first minor issue (Cardiac Amyloidosis is reported with two different acronyms: please check and address). I found again indicated in two different ways. In particular at line 361 they named AC: I guess it stands for Cardiac Amyloidosis but it is counfounding.
Author Response
"Please see the attachment."

Reviewer 3 Report
The athos have successfully addressed the reviewers comments. I does not have further remarks for the revised manuscript.
